# A Blockchain-Based OCF Firmware Update for IoT Devices †

**Elizabeth Nathania Witanto** [1] , **Yustus Eko Oktian** [1] , **Sang-Gon Lee** [1,*] and **Jin-Heung Lee** [2]

1   College of Software Convergence, Dongseo University, Busan 47011, Korea;
    elizabethnathaniaw93@gmail.com (E.N.W.); yustus.oktian@gmail.com (Y.E.O.)
2   Daun Information and Communication, Busan 48058, Korea; leejinheung@gmail.com
*   Correspondence: nok60@dongseo.ac.kr
†   This work is an extended and revised version of our conference paper that was presented in ICTC2019:
    Witanto, E.N.; Oktian, Y.E.; Kumi, S.; Lee, S.G. Blockchain-based OCF Firmware Update. In Proceedings of
    the 2019 International Conference on Information and Communication Technology Convergence (ICTC),
    IEEE, Jeju Island, South Korea, 16–18 October 2019; pp. 1248–1253.

**Abstract:** As the usage growth rate of Internet of Things (IoT) devices is increasing, various issues related to these devices need attention. One of them is the distribution of the IoT firmware update. The IoT devices' software development does not end when the manufacturer sells the devices to the market. It still needs to be kept updated to prevent cyber-attacks. The commonly used firmware update process, over-the-air (OTA), mostly happens in a centralized way, in which the IoT devices directly download the firmware update from the manufacturer's server. This central architecture makes the manufacturer's server vulnerable to single-point-of-failure and latency issues that can delay critical patches from being applied to vulnerable devices. The Open Connectivity Foundation (OCF) is one organization contributing to providing interoperability services for IoT devices. In one of their subject areas, they provide a firmware update protocol for IoT devices. However, their firmware update process does not ensure the integrity and security of the patches. In this paper, we propose a blockchain-based OCF firmware update for IoT devices. Specifically, we introduce two types of firmware update protocol, direct and peer-to-peer updates, integrated into OCF firmware update specifications. In the direct scenario, the device, through the IoT gateway, can download the new firmware update from the manufacturer's server. Meanwhile, in the peer-to-peer scheme, the device can query the update from the nearby gateways. We implemented our protocol using Raspberry Pi hardware and Ethereum-based blockchain with the smart contracts to record metadata of the manufacturer's firmware updates. We evaluated the proposed system's performance by measuring the average throughput, the latency, and the firmware update distribution's duration. The analysis results indicate that our proposal can deliver firmware updates in a reasonable duration, with the peer-to-peer version having a faster completion time than the direct one.

**Keywords:** blockchain; firmware update; IoT; IoTivity; OCF

## 1. Introduction

The center of a notable technology shift in our world is the Internet of Things (IoT). The IoT allows several sensor devices to connect to the Internet and share IoT data. By now, around 14 billion of "things" are connected to the Internet. A forecast from International Data Corporation (IDC) estimates that there will be 41.6 billion connected IoT devices in 2025 [1]. With this amount of IoT devices, the upcoming issue is interoperability between IoT devices from different manufacturers.

The IoT device software development does not end even when the manufacturer sells the device to the customer. The manufacturer still needs to update the firmware periodically for several reasons.

1. Releasing new features through the firmware updates.
2. When manufacturers find bugs from their released products, the firmware updates can facilitate the distribution of bug fixes.
3. As the hackers try to exploit security vulnerabilities continuously, the manufacturers need to regularly release updates to prevent cyber-attacks. That final reason is the most crucial one, as it may cause catastrophic damage. For instance, attackers can manipulate the self-driving car's parameters to perform brake failures that may inflict dangerous accidents.

Based on several reasons mentioned above, the manufacturers need to ensure that the firmware updates are available on time and are safe for the IoT devices. Nowadays, we commonly depend on the over-the-air (OTA) updates for IoT devices because they are quick and convenient for users [2]. To update a particular device, users do not need to bring their device to the manufacturer. Instead, they can connect their device to a computer and download the updates through the Internet. However, since most OTA updates happen in a centralized way, they are vulnerable to single-point-of-failure and latency issues. Server failures and extended network latency can delay critical patches from being applied to exposed IoT devices. Moreover, the firmware update process must also guarantee the integrity of the patch file.

Open Connectivity Foundation (OCF), as one of the organizations that supports IoT devices' interoperability, provides a firmware update protocol [3,4]. However, their protocol does not have security protection because they only discuss simple check-new-update interactions between the OCF device and the manufacturer. Specifically, there is no step to ensure the firmware update integrity in the OCF protocol. When the adversary intercepts the firmware update file transmission and replaces it with malicious malware, the device will install the malware instead of the original update file. Therefore, it is essential to add more security steps in the OCF protocol.

Blockchain is a promising technology to solve integrity and trust issues. In particular, it is challenging to modify the information once it enters the blockchain. This feature makes the blockchain an immutable system. Moreover, the blockchain keeps synchronous digital transaction records in each participant's local storage. All members can then take part in monitoring and verifying the records. Thus, this property enhances trust in the system.

In this paper, we propose a blockchain-based OCF firmware update for IoT devices. The manufacturer can use the blockchain to store the metadata of IoT devices' updates. When an IoT device receives a firmware update file, it can confirm whether metadata matches the metadata previously saved in the blockchain. If they are not identical, then a malicious alteration must exist on the update file. Moreover, in our proposal, IoT gateways can share the downloaded firmware updates from the manufacturer's server. As a result, we can disseminate traffic burden from the server to gateways. Therefore, our proposal can avoid a single-point-of-failure issue and enhance the overall firmware update availability.

In summary, we make the following contributions:

- We integrate blockchain and smart contracts to the OCF firmware update protocol to enhance its integrity and transparency. The manufacturers use smart contracts in the blockchain network for firmware update metadata storage. As the blockchain provides a hard-to-tamper reference, it can prevent malicious changes from adversaries.
- We improve the OCF update availability by giving two options for distributing the update. In the direct firmware update distribution, the IoT devices immediately download the update from the manufacturer's server. Meanwhile, in the peer-to-peer scheme, the device can query the update from the nearby gateways. With these two ways of distributing the update, we increase the overall robustness of the update process.

We organize the rest of the paper as follows. Section 2 provides a brief introduction to the blockchain, Ethereum, smart contracts, OCF, and IoTivity. Section 3 summarizes previous work of the

existing method for the blockchain-based firmware updates in general cases. We describe the proposed architecture and protocol design in Section 4. After that, we explain the details of our implementation in Section 5 and evaluate it in Section 6. Finally, we conclude in Section 7.

## 2. Background Theory

### 2.1. Blockchain

Blockchain is a distributed database of records or a public ledger of digital transactions that have been executed and shared between several participants [5]. Anyone in the blockchain network can monitor and verify the authenticity of the transactions. The consensus of a majority of the participants in the system will verify each transaction in the public ledger. Blockchain creates permanent records and histories of transactions. However, the permanence of the record is based on the permanence of the network. In the context of blockchain, this means that a large portion of a blockchain community would all have to agree to change the information and is incentivized not to change the data [5]. For example, in the blockchain, the attacker needs to control more than 50% of the network nodes to make the blockchain network compromised [6]. However, it is difficult and costly to have that amount of computing power at this time.

The blockchain technology relies on two cryptographic methods, which are digital signatures and cryptographic hash functions. The digital signature provides the authenticity of a digital message. In the blockchain, it can be used to provide integrity, authentication, and non-repudiation. There are two properties required for the digital signature scheme [7]. The first property is straightforward; it means that valid signatures must be verified. For example, if someone signs a message with their secret key ($sk$), and his/her friend later tries to validate that signature over that same message using his/her public key ($pk$), the signature must validate correctly. The second requirement is that it is computationally infeasible to forge signatures. An adversary who knows your public key and gets to see your signatures on some other messages cannot forge your signature on some message for which he has not seen your signature.

The cryptographic hash function will compute a hash value from the given input. In the blockchain, each block will store the hash value of the previous block. The change of one datum can be easily detected. If the adversary changes any data of one block, the next block's data will be changed. Therefore, the cryptographic hash function has an essential role in maintaining data integrity in the blockchain network. This function should have four properties:

1.  Deterministic means that the same input will always produce the same output.
2.  One-way function means that the function can easily compute the input, but it is infeasible to revert it from the hash value [8].
3.  Collision resistance is when it is difficult to find two different messages as input with the same hash value, for instance, $m_1$ and $m_2$, and $hash(m_1) = hash(m_2)$ [9].
4.  Pseudorandom means that a change in the input must result in an unforeseeable change of the output. If the hash value of the input two was four, the hash of three better not be six [10].

### 2.2. Ethereum and Smart Contracts

One of the well-known blockchain protocols is Ethereum. Ethereum is an open-source, public, and blockchain-based distributed ledger featuring smart contracts functionality. It enables developers to build blockchain applications with business logic that executes in a decentralized environment while leveraging the high availability of the Ethereum network [11]. There are two different types of accounts in Ethereum: externally owned accounts (EOAs) and contract accounts. Users control EOAs, often via software such as a wallet application external to the Ethereum platform.

In contrast, contract accounts are controlled by program code (also commonly referred to as "smart contracts") that is executed by the Ethereum Virtual Machine [12]. Nick Szabo created the term "smart contract" in the 90s. Szabo used a vending machine's basic example to describe how real-world

contractual obligations can be programmed into software and hardware systems. Everyone who puts the correct amount of coins into the machine can expect to receive a product in exchange [13]. Similarly, on Ethereum, contracts can hold value and unlock it only if specific conditions are met. It is defined as computer code running on top of a blockchain and is correctly executed without fraud or any interference from a third party [13]. The owner publishes a smart contract, and it will exist at an address (its public key) in the network. The users and the contract owner can interact with the contract by calling the already defined functions. The contract is transparent and deterministic—its behavior is defined at the time of its publication, and its code is available for inspection on the blockchain [14]. Additionally, a smart contract is immutable. After deploying the code, a smart contract cannot be changed. The only way to modify the smart contract is to deploy a new instance [12].

### 2.3. Open Connectivity Foundation (OCF) and IoTivity

The OCF is one of the organizations that contribute to the IoT standardization to provide interoperability. The OCF is also an industry group that delivers an IoT software framework specification and a product certification program. The organization includes well-known companies such as Intel, Cisco, General Electric, Samsung, Microsoft, Qualcomm, and IBM in its membership. The OCF specification has adopted a resource-oriented architecture (ROA), implying that information and concepts are represented as resources. Resources are specified in RESTful API Modelling Language (RAML). Each resource definition contains a unique identifier, an indication of whether the resource is an actuator or a sensor or another type, and a list of supported methods a JSON schema for input and output for each method [3,15].

IoTivity is an open-source framework that implements the OCF protocol for easy and secure device-to-device communication for IoT devices [16,17]. It runs as middleware and aims to provide connectivity devices to the cloud and device to device. The software framework is licensed under the Apache license version 2.0.

### 2.4. OCF Firmware Update Protocol

As shown in Figure 1, the OCF firmware update protocol is triggered by the client's update request. For the convenient explanation of our proposed protocol, we add two notes with black circle mark (●) that are called "new firmware available" and "software validation" into the original OCF firmware update protocol. The entities involved in this protocol are the client, `pstat`, `softwareupgrade`, and external server. The client is the OCF device that will initiate the update. The `pstat` and `softwareupgrade` are the OCF resource model. They reside in the OCF device. The resource model defines the concepts and mechanisms that provide consistency and core interoperability between OCF ecosystems devices [3,4].

The `softwareupgrade` resource is used to control software updates of the device [3,18]. Then, the `pstat` resource stands for provisioning status resource. It maintains the device's provisioning status [19,20]. The `softwareupgrade` resource has a property value called `state`. It will show idle, new software available (NSA), software version validation (SVV), software version available (SVA), and upgrading value. This `state` will affect the `pstat`'s property value. The `pstat` resource has several property values, but there are two properties that will take part on this case—target mode (`tm`) and current mode (`cm`). The value of this property will be changed along with the `softwareupgrade` `state`'s value. The values of `tm` and `cm` will be binary, where the `tm` bit indicates that the action is initiated and the `cm` bit indicates the result of the action [19,20].

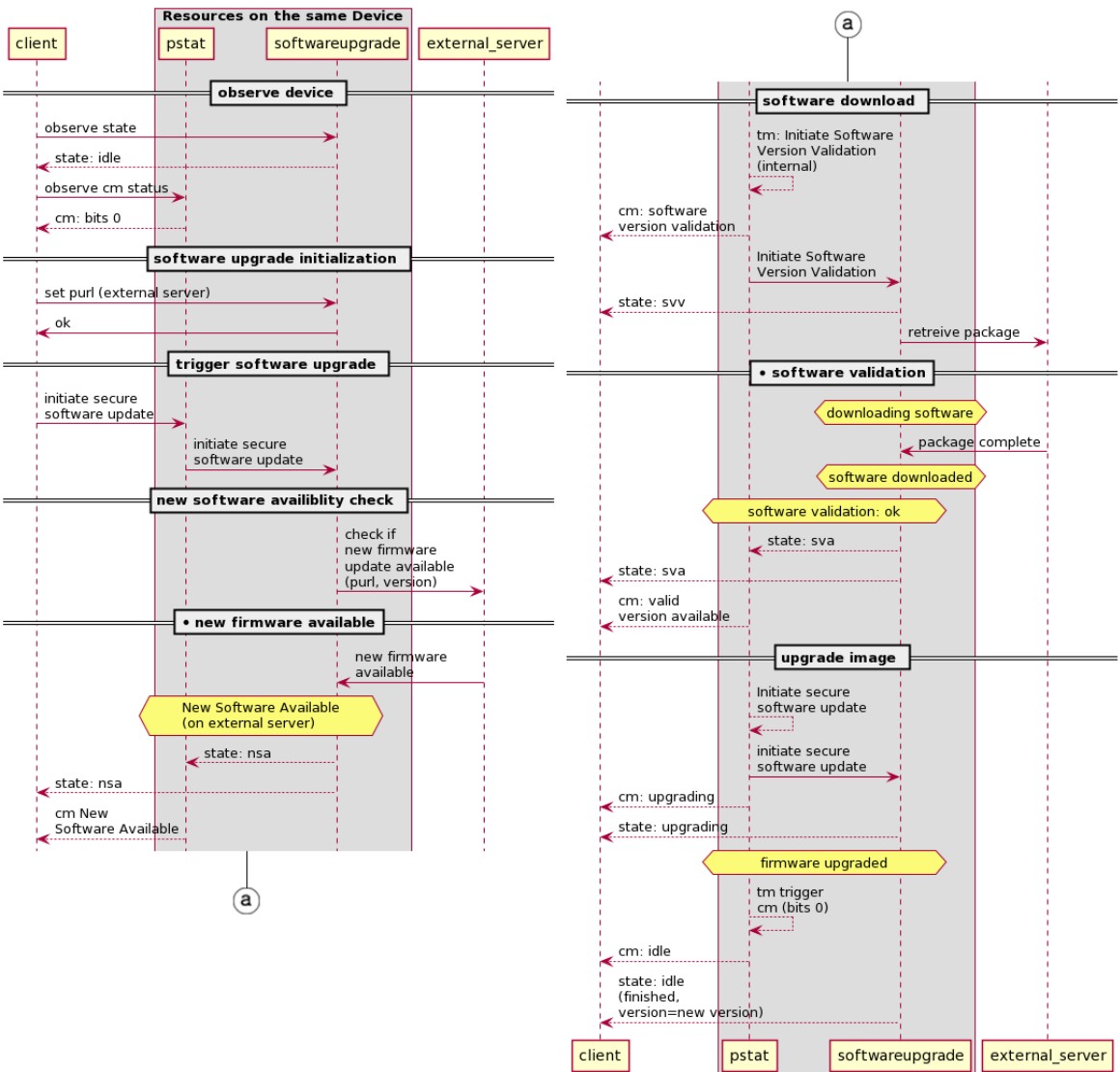

**Figure 1.** OCF firmware update protocol, adapted from [3] (from top-left to bottom-right).

From the "observe device" step to the "trigger software update" step, the OCF device will check the `state` of `pstat` and `softwareupgrade` resources. If these two resources are idle, the OCF device will initiate the request to check the firmware update on the external server in the "new software availability check" step. On receiving notification of firmware update is available from the server, the `pstat` and `softwareupgrade` resource will set their `state` to new software available. At the end of the "software download" step, it will retrieve the package from the external server. In the "software validation" and "upgrade image" steps, those two OCF resources will set the `state` to downloading and installing the new firmware update. A more detailed explanation is provided in Section 4.

## 3. Related Work

In [3,4], OCF provides a specification for the firmware update process from the OCF device to the manufacturer (external server). However, their protocol does not have security protection because they only discuss simple check-new-update interactions between the OCF device and the external server. Specifically, there is no step to ensure the integrity of the firmware update in the OCF protocol. To ensure the update file's integrity, we can simply modify the protocol by including checking of the firmware package signatures. However, we introduced blockchain in the OCF firmware update protocol. By using blockchain technology, peer-to-peer firmware update becomes possible. Thus,

IoT devices can share the available newest update through IoT gateways. Then, they can check the integrity and validity of the update to the blockchain network. As the gateways provide update distribution on behalf of the manufacturer, single-point-of-failure is prevented, and firmware update availability is increased.

Our proposed protocol is specific for the OCF firmware update process. However, we also compare our proposed protocol to the other paper about the general IoT firmware update that utilizes blockchain. Lee and Lee [21] proposed a blockchain-based secure firmware update for embedded devices in an IoT environment. This method gives some possibility to check the firmware version before the update begins, which one has a higher or lower version. However, this will later give redundancy checking that may cause unnecessary network traffic and computational power consumption and operations for the nodes. Besides, the method only provides showcases for firmware updates from one manufacturer instead of the various manufacturers. In fact, many IoT devices from various manufacturers will exist in an IoT environment. Therefore, this firmware update framework does not apply to a heterogeneous IoT environment. Boudguiga et al. proposed the other scheme of blockchain technology for firmware update [22]. There are innocuousness checking nodes that acknowledge the transaction before the IoT device can download and install it. Therefore, it delays the update operation.

These previous two solutions use Bitcoin technology. In Bitcoin, it takes around 10 min to verify a transaction and create a new block. For Ethereum, the average time for one block creation is 10–20 s [23]. Time is an essential factor when distributing a newer version of firmware to the IoT devices. Moreover, developers can utilize smart contracts in Ethereum to build a decentralized application because smart contracts will execute when the conditions are met. In the case of firmware updates' distribution, each version of firmware released by the device's manufacturer will be recorded in the smart contracts. When devices want to download the new firmware, they can verify it through the smart contracts. Furthermore, Ethereum provides a lightweight node for constrained and low capacity devices that cannot afford to store multiple dozen Gigabytes of blockchain data [24,25].

Yohan et al. [26] also proposed a firmware update protocol based on blockchain. Their protocol was designed based on Ethereum. They provide direct and indirect firmware update distribution. However, in their scheme, they process the received messages without verification, which means that there is a possibility that the adversary can send the messages, or the adversary can modify the messages.

## 4. The Proposed Protocol

In this section, we propose blockchain-based OCF firmware update protocols. The update protocols consist of two ways of distributing firmware updates: a direct and a peer-to-peer update. In the direct firmware update, the device, through an IoT gateway, can download the new firmware update directly from the manufacturer's server. Meanwhile, in the peer-to-peer scheme, the device can query the update from the nearby gateways that have downloaded the new firmware. By providing a peer-to-peer update scheme, because the gateways provide update distribution on behalf of the manufacturer, we can prevent a single-point-of-failure and increase overall firmware update availability. Moreover, Ref. [22] has innocuousness checking nodes that check the update against bugs and known attacks every time the manufacturer releases an update. Nevertheless, in our proposed scheme, before the manufacturer releases a new firmware update, a trusted party (such as certified cybersecurity companies) must already have checked it. This process will shorten the firmware update's distribution time. Protocols in [26] provide two types of blockchain-based firmware update protocols. One is initiated by the manufacturer's server and another initiated by the IoT device. The OCF protocol provides the IoT device initiated protocol. Thus, we implement only the IoT device initiated protocol. The components in protocol [26] send the messages without any integrity protection. However, our proposed protocol provides the messages integrity protection with a digital signature and hash algorithm. That is crucial because it will avoid allowing the adversary to alter the messages and inject malicious code during transmission.

### 4.1. The Architecture

As shown in Figure 2, three entities are involved in our protocol; the explanations of the components follow.

- *Manufacturer (full node).* The manufacturers of the IoT devices such as smart bulb, thermostat, or the other. It stores and releases a new firmware update for their devices. This entity is also a node in the blockchain network. It will publish the firmware update metadata to the smart contracts in the blockchain network. The set of manufacturers $M$ is denoted as:

$$M = \{m_1, m_2, ...m_n\}$$

- *IoT gateway (lightweight node).* The IoT gateway can be a Wi-Fi router in a smart home or another device. The information of the IoT devices will be stored in the IoT gateway (e.g., device manufacturer, device type, latest firmware version). The IoT gateway will download the new firmware update for the IoT device from the manufacturer or from another gateway. The IoT gateway is also a blockchain node. When the manufacturer stores the firmware update metadata to the smart contracts, it will receive a notification and save the information to the local database. The set of IoT gateways $G$ is denoted as:

$$G = \{g_1, g_2, ...g_n\}$$

- *IoT devices.* The sensors or embedded devices. In our cases, it will be the OCF devices. IoT device will request and download the new firmware update from the IoT gateway. The set of IoT devices $I$ is denoted as:

$$I = \{i_1, i_2, ...i_n\}$$

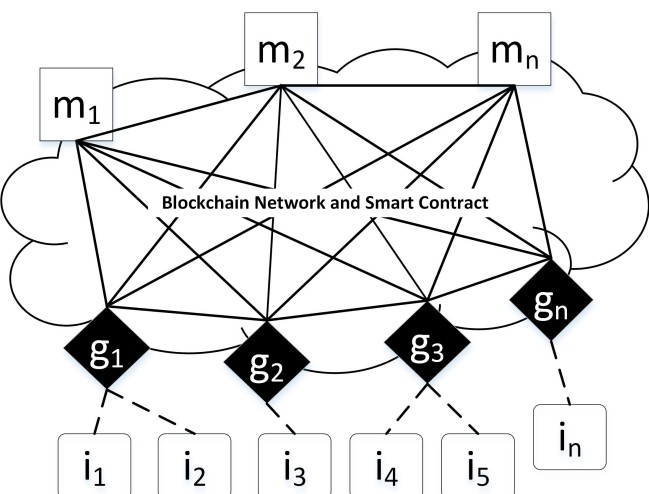

**Figure 2.** Our proposed blockchain-based firmware update architecture.

### 4.2. The Protocol Design

Our proposed protocol has three parts: the creation of the smart contract, direct firmware update distribution, and peer-to-peer firmware update. Table 1 shows the notation that we use in our design protocol. Additionally, in our protocol, the OCF device sends or receives a request from the IoT gateway instead of directly to the external server or the manufacturer.

**Table 1.** The notation in the proposed protocol.

| Name | Description |
|---|---|
| $U_{id}$ | The update identifier |
| $U_{bin}$ | Binary file of the new firmware update |
| $f_v$ | Newest firmware version |
| $M_{id}$ | The manufacturer identifier |
| $dtype$ | The device type |
| $purl$ | The package URL of the firmware update file |
| $SK_m$ | Manufacturer's secret key |
| $SK_g$ | IoT gateway's secret key |
| $M_{sc}$ | Manufacturer's smart contract |

### 4.2.1. Creation of the Smart Contract

Figure 3 shows that the manufacturer releases a new firmware update. It stores the hash of update $U = (U_{id}, f_v, M_{id}, dtype, purl)$ that is signed with the manufacturer's secret key ($SK_m$) to the smart contracts. $U_{id}$ is the update identifier. It is simply the hash of the firmware update's binary file. The smart contract will verify this request. If it is valid, the update $U$ will be stored to the smart contract ($M_{sc}$), and it will be published to the blockchain network. Before the manufacturer releases a new firmware update, the trusted party (for example, by certified cybersecurity companies) must already check the new firmware update. They will check the firmware update as to whether it is free of bugs and malicious attacks. In our protocol, since the IoT gateway is a blockchain node, it will receive a notification when the manufacturer stores the metadata to the smart contract.

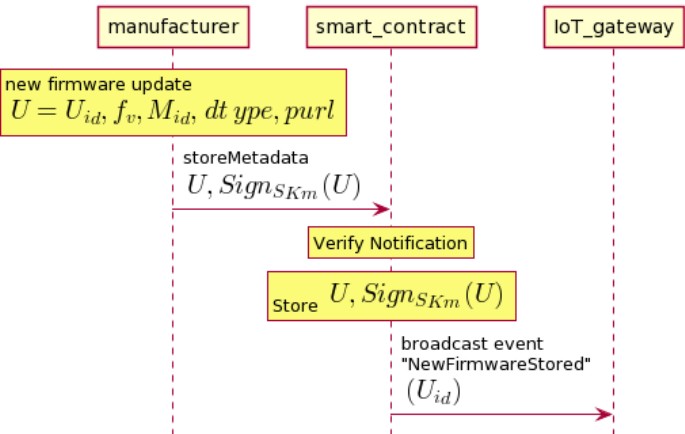

**Figure 3.** Creation of the firmware update smart contract.

### 4.2.2. Direct Firmware Update Distribution

Four entities are involved in this protocol. They are the OCF device, IoT gateway, smart contract, and manufacturer. As shown in Figure 4, the distribution begins when the client initiates the update process to the OCF device. We put the detailed explanation of the direct firmware update protocol below:

- *OCF device: initiate the firmware update.* First, the client will observe the `state` of `softwareupgrade` and `pstat` resource—whether it is idle or not. If `softwareupgrade` resource's `state` is idle and `pstat` resource's `cm` value is 0, the client will set the package URL (*purl*) in the `softwareupgrade`. This action will trigger the firmware update process by setting the `pstat state`'s value to "initiate secure software update." Then, `pstat` will forward it to the `softwareupgrade`. After the `state` of `softwareupgrade` and `pstat` show that it is ready to continue the process, the `softwareupgrade` resource will check if there is new firmware update by sending the *purl* and the current firmware version as parameters to the IoT gateway.

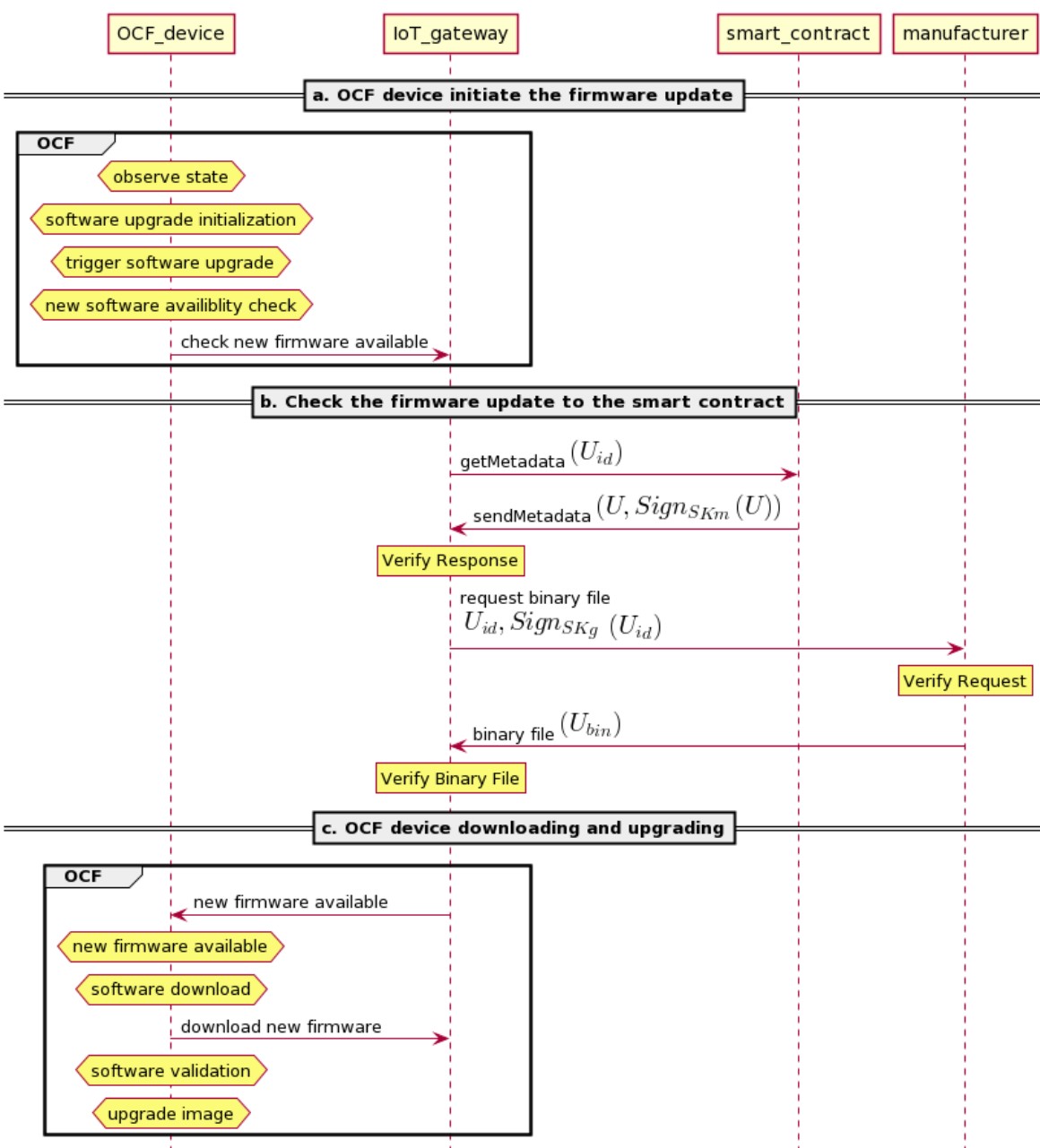

**Figure 4.** Direct firmware update distribution: gateway downloading a new firmware update from the manufacturer.

- *Check the firmware update to the smart contract and download the firmware update from the manufacturer.* After receiving an update request from the OCF device, the IoT gateway will check the newest firmware update to the smart contract ($M_{sc}$) for this particular device by sending the $U_{id}$. If there is a firmware update contract for the device, $M_{sc}$ will reply with the signed U that is stored in the contract to the IoT gateway. The IoT gateway will receive this message and verify the signature. If valid, it will send the signed $U_{id}$ to the manufacturer to request the binary file. Then, the manufacturer will verify the signature. If verified, it will reply with the binary file ($U_{bin}$) to the IoT gateway. The IoT gateway will verify the binary file by comparing the hash value of $U_{bin}$ and the $U_{id}$ value from the manufacturer. If they are equal, the IoT gateway will send a notification to the OCF device.

- *OCF device: downloading and upgrading.* On the OCF device's side, as shown in Figure 1, the `softwareupgrade` resource will notify the `pstat` resource and the client of a new software update by sending NSA `state`. The `pstat` will set the `cm` value to "new software available" and notify the client. Moreover, the `pstat` will set its internal state of `tm` to "initiate software version validation." Later, the `pstat` will relay this state to the client and change the `cm` value to "software version validation." Subsequently, the `pstat` will notify the state's changing to `softwareupgrade`. Then, the `softwareupgrade` will inform the state's changing to the client by sending SVV as its `state` value. In the next step, the OCF IoT device will send "retrieve the package" request to the IoT gateway through the `softwareupgrade` resource. After the package has been downloaded, the `softwareupgrade` resource will send SVA state to the `pstat` resource and client. The `pstat` will set the `cm` value to "valid version available." Afterward, it will set its internal "initiate secure software update" `state` and send the same `state` to the `softwareupgrade`. In order to upgrade the firmware, the `pstat` and `softwareupgrade` will send the upgrading `state` to the client. After successfully installing the new firmware, the `pstat` will change the `tm` bit value to 0. Then the `pstat` and `softwareupgrade` will notify their idle state to the client.

The proposed protocol provides an integrity service to the firmware update by using the signature/verification scheme on the firmware update checking step.

### 4.2.3. Peer-To-Peer Firmware Update Distribution

The peer-to-peer firmware update protocol happens between devices with the same specifications and from the same manufacturer. The device's gateway will download from the other gateway that has been downloaded the new firmware. As shown in Figure 5, IoT gateway 2 requests an update from IoT gateway 1.

- *OCF device initiating the firmware update.* For this part, see the explanation in Section 4.2.2.
- *Update availability check.* As shown in Figure 5, after receiving an update request from the OCF device $i_2$, IoT gateway 2 will check the newest firmware update to the smart contract ($M_{sc}$) for this particular device by sending the $U_{id}$. After that, $M_{sc}$ will check the contract for compatibility with the request. If there is a firmware update contract for the device, $M_{sc}$ will send the signed U stored in the contract. IoT gateway 2 will receive this message and verify the signature. If valid, it will send a request to IoT gateway 1 to check the availability of the firmware. IoT gateway 1 will search for the compatible update for this device in the local database. If available, it will send the signed U and challenge c to give the update notification; otherwise, it will send a signed NA (not available) message. After that, IoT gateway 2 will verify the signature. If verified and the message is not NA, it will continue; otherwise, the process will be terminated. The next step is comparing the $U_{id}$ from IoT gateway 1 and $U_{id}$ from $M_{sc}$. If the value is equal, IoT gateway 2 will create a connection to IoT gateway 1 for downloading the firmware. Afterward, IoT gateway 2 can verify the binary file by comparing the hash of $U_{bin}$ from the IoT gateway 1 and the $U_{id}$ value. If the value is equal, IoT gateway 2 will forward it to the OCF device; otherwise, it will reject it.
- *OCF device downloading and upgrading.* For this part, see the explanation in Section 4.2.2.

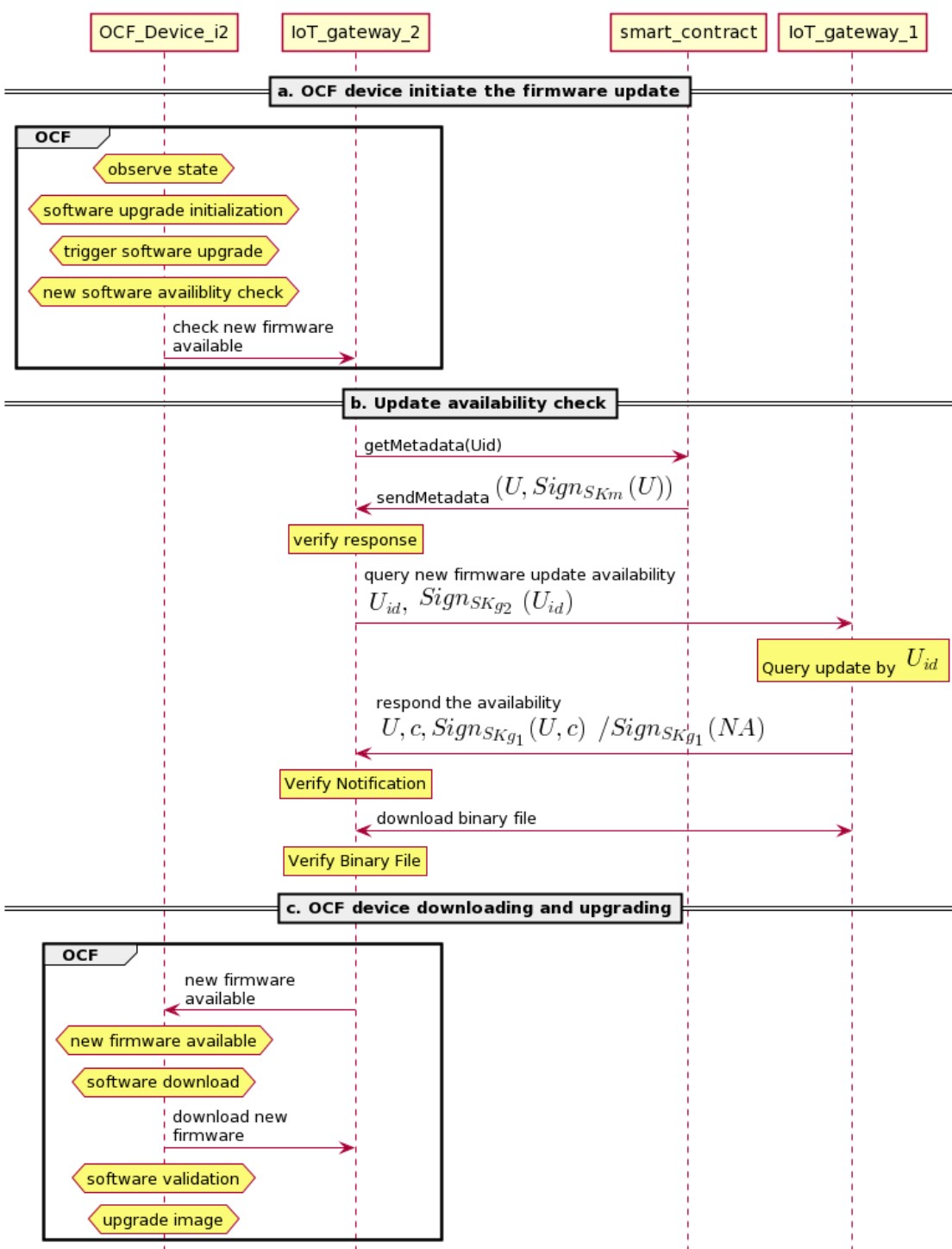

**Figure 5.** Peer-to-peer firmware update distribution: gateway downloading a new firmware update from another gateway.

Like the direct update protocol, the peer-to-peer update protocol also provides the integrity service to the firmware update.

## 5. Implementation

### 5.1. Development Environment

Our proposed method's development environments are summarized in two tables, one for the manufacturer and the blockchain network, and another for the IoT gateway and IoT device, as shown in Tables 2 and 3. Both manufacturer and blockchain network reside in Ubuntu Linux 16.04 LTS under the VMWare virtual environment. The virtual environment was installed on a Windows 10 host machine with Intel Core i5-7200U @2.50 GHz CPU and 4 GB memory. For the manufacturer, we used Express.js to generate a REST API. It provides GET requests for IoT gateways. Web3.js was used to facilitate communication between the manufacturer and the blockchain network. Additionally, we used Node.js as a programming language. Our blockchain network was based on Ganache. Ganache is a rapid Ethereum and Corda distributed application development [27]. Solidity and Node.js were used as programming languages to write a smart contract.

**Table 2.** Development environment for the manufacturer and the blockchain network.

| Parameters | Manufacturer | Blockchain Network |
|---|---|---|
| CPU | Intel Core i5-7200U @2.50 GHz | |
| Memory | 4 GB | |
| OS | Ubuntu Linux 16.04 LTS | |
| Node | v.13.14.0 | |
| Ethereum client | x | Ganache-client |
| Library and Framework | Express.js, Web3.js | x |
| Programming language | Node.js | Solidity, Node.js |

Table 3 describes the development tools and technologies for implementing the IoT gateway and the IoT device. Raspbian Buster was installed on the Raspberry Pi 3 Model B with 1 GB memory for both the gateway and the IoT device. The gateway used SQLite3 as a database management system to store device information and firmware updates' metadata from smart contracts. Express.js was used to generate REST API requests (POST and GET requests) and Web3.js to enable communication between the IoT gateway and the blockchain network. Additionally, Node.js was used as a programming language for both the gateway and the IoT device. Nevertheless, the IoT device had C as an additional programming language for IoTivity's code. Besides, IoTivity Lite 2.0.5 was installed to make this device an IoTivity server. Additionally, the IoT device was connected to the Dallas temperature sensor.

**Table 3.** Development environment for the IoT gateway and the IoT device.

| Parameters | IoT Gateway | IoT Device |
|---|---|---|
| Hardware | Raspberry Pi 3 Model B | |
| Memory | 1 GB | |
| OS | Raspbian Buster | |
| Database | SQLite3 | x |
| Server | x | IoTivity Lite 2.0.5 |
| Resources | x | Temperature |
| Library and Framework | Express.js, Web3.js | Express.js, LibCurl |
| Programming language | Node.js | Node.js, C |

### 5.2. Implementation Design

The proposed method's use case deployment is divided into three parts: implementation of the creation of firmware update smart contract, direct firmware update, and peer-to-peer firmware update distribution. In our implementation, there were five entities. We used the VMWare virtual environment to set up the manufacturer and blockchain network. Our blockchain network was based on Ganache. The IoT gateway and IoT device were hosted on Raspberry Pi. In the IoT gateway,

SQLite3 was used as a database to store metadata from smart contracts and device information. For handling the REST API request, we used Express.js. It supports GET requests for IoT devices and manufacturers. To communicate with the blockchain network, Web3.js was used in the IoT gateway and the manufacturer. The IoT device was connected to a physical temperature sensor. Besides, it hosted the IoTivity server while the client hosted the IoTivity client. All of these entities were in the same network. Figure 6 shows our implementation of IoT network setup. We used a Wi-Fi link to simulate a remote connection to the manufacturer and used a wired link to simulate the local connection between gateways.

### 5.2.1. Creation of the Smart Contract

First, we will explain the implementation of the creation of firmware update smart contract of Figure 3. As shown in Figure 6, the manufacturer has a new firmware update in the software update repository. To distribute the new firmware, the manufacturer stores the metadata to the smart contract. By the end of this process, the IoT gateways that participate in the blockchain network will be able to obtain the new firmware update information and save it to its local database. The detail explanation is described as follows:

1. The manufacturer creates a firmware update smart contract.
2. The manufacturer stores the metadata ($U$) to the smart contract. The metadata consist of $U_{id}, f_v, M_{id}, dtype, purl$ which are signed with the manufacturer's secret key ($SK_m$).
3. "NewFirmwareStored" event in the blockchain network is triggered. Since the IoT gateways is a blockchain node, it will receive the event notification. The $U_{id}$ is sent through this event.
4. The IoT gateways call the "getMetadata" function in the smart contract to request the full metadata.
5. The IoT gateways receive the metadata ($U$).
6. The IoT gateways verify the signature. We use the "recover" function from EthCrypto's library to recover the signer's public address in the blockchain network with their signature as input [28]. If verified, it will save the metadata to the local database; otherwise, the data are rejected.

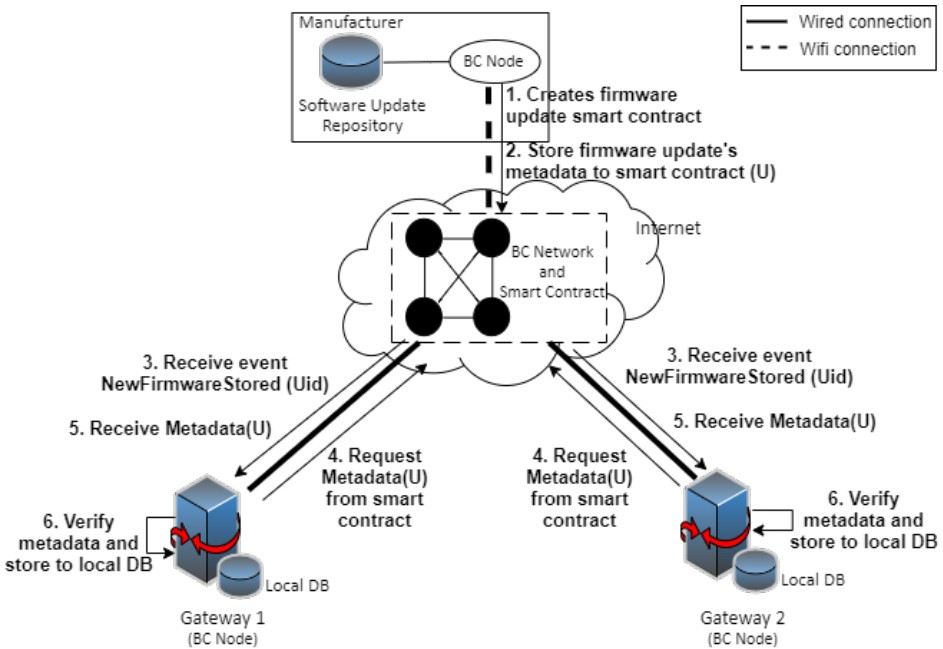

**Figure 6.** Implementation of creation of firmware update smart contract.

5.2.2. Direct Firmware Update Distribution

The second part is the direct firmware update distribution of Figure 4. As shown in Figure 7, direct firmware update distribution starts when the client sends an update request (step 1). According to OCF firmware update protocol, their protocol is a scheduled firmware update process [3,4]. Thus, in the IoTivity Lite GitHub [29], the client will need to send a POST request to the IoTivity server that resides in the IoT device with three parameters—package URL (*purl*), software update's action (*swupdateaction*), and the time to update (*updatetime*). The package URL will be directed to the IoT gateway address, the *swupdateaction* will be set to "*isac*" (initiate secure availability check), and the *updatetime* will be set to the time when the client wants the IoT device to update the firmware with the format "YYYY-MM-DDThh:mm:ssZ." According to the scheduled time, the *swupdateaction* will be changed to "*isac*." Then, it will trigger the next step of the firmware update process. Three functions exist in the IoTivity server to process the firmware update based on the step. The names of the functions are Check New Version, Download Update, and Perform Upgrade. Therefore, we created three REST APIs to utilize these functions to communicate with the IoT gateway and install the new firmware update. We explain more about three functions below:

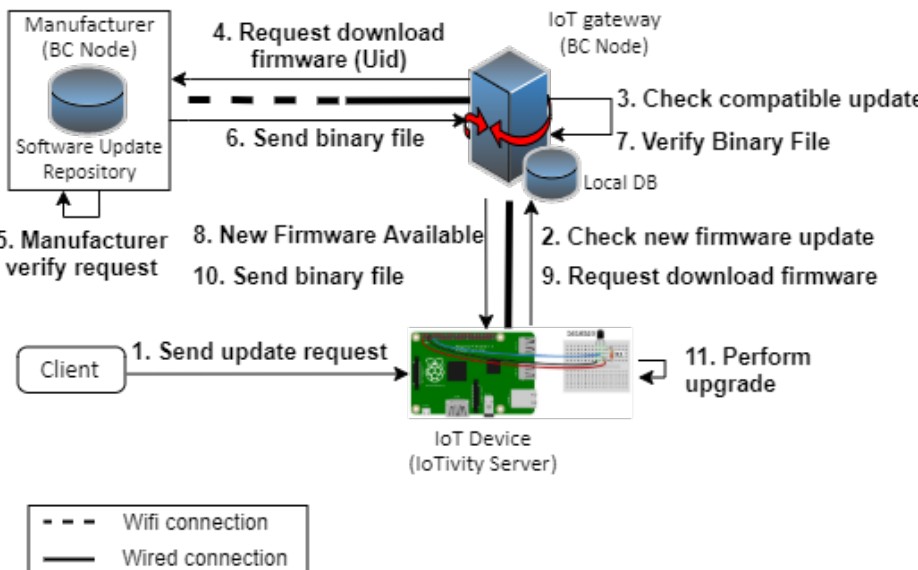

**Figure 7.** Implementation of a direct firmware update distribution.

- **Check New Version (step 2–8)**. If it is time to update, the IoT device will request the IoT gateway to check for a new update (2). The IoT gateway will then query the local database as to whether there is any compatible update based on the manufacturer and the device specifications (3). Additionally, it will check whether the current firmware version is lower than the value in the database. If the checking conditions are all matched, the IoT gateway will send $U_{id}$ and firmware version to the manufacturer and sign it with the gateway's secret key ($SK_g$) (4). On the manufacturer side, after receiving the request, it will verify the signature using EthCrypto.recover (5). If verified, it will connect to the IoT gateway to deliver the binary file (6). After the IoT gateway receives the binary file, it will verify the hash of the binary file and compare the hash with the $U_{id}$ (7). If they are equal, a notification will be sent to the IoT device, and otherwise reject it (8).
- **Download Update (step 9–10)**. After receiving a notification, the IoTivity server in the IoT device will change the state of pstat and softwareupgrade resources. It will then request to download the binary file to the IoT gateway (9). After receiving the request, the IoT gateway will send the binary file to the IoT device (10).

- **Perform Upgrade (step 11)**. In this step, the IoT device extracts and installs the new firmware. To proceed with this step, we utilize Diff3 and patch function. Before we explain further, we will explain the process on the manufacturer's side. Thus, in our implementation, when the manufacturer has a new firmware update, they use Diff3 to compare old and new firmware files and then to merge them into one new file with .patch format. Figure 8 shows an example of a .patch file. The manufacturer wants to add a new function so that the IoT device can read the temperature in Fahrenheit. In the old version, the IoT device only read in Celsius. This file will be used to update the old firmware in the IoT device. Diff3 makes the update faster. It will not replace the whole file, but it will only remove or add the lines as in the written .patch file. After creating the .patch file, the manufacturer will create a MAKE file with commands in it to execute the patch. Then, the manufacturer will insert both .patch and MAKE files into an archive (.zip) file. After the IoT device has the .zip file, it will extract the .zip file and run the MAKE file to start the .patch file.

**Figure 8.** The .patch file created using Diff3.

### 5.2.3. Peer-To-Peer Firmware Update Distribution

As shown in Figure 9, the implementation of a peer-to-peer firmware update consists of four entities: the client, the IoT device (IoTivity server), IoT gateway 1, and IoT gateway 2. Similarly to the direct firmware update protocol, when the client wants to update the firmware, the client needs to set the time, the action, and the URL to the IoTivity server that resides in the IoT device. We also implemented three REST APIs to support communication between IoTivity server and IoT gateway: Check New Version, Download Update, and Perform Upgrade. The flow is the same as we explained in Section 5.2.2, except, in this case, IoT gateway 1 has the new firmware update on its local database, and IoT gateway 2 does not have the new firmware update. Instead of the manufacturer, IoT gateway 2 will download the new firmware from IoT gateway 1. Since they are in the same network and each gateway has an IP address list, IoT gateway 2 will send a broadcast message to the IP address list to determine which one has the new firmware update. After finding the gateway with the newest firmware update, IoT gateway 2 will download the update from this gateway instead of the manufacturer. Thus, in the peer-to-peer update, the primary interaction is between the gateways. It will not include the manufacturer as the host to download the firmware update's file. Since gateways help the firmware update distribution jobs, it will help to avoid the single-point-of-failure problem and increase overall firmware update availability. Additionally, it maintains the firmware update file's reliability by using a smart contract in the blockchain network.

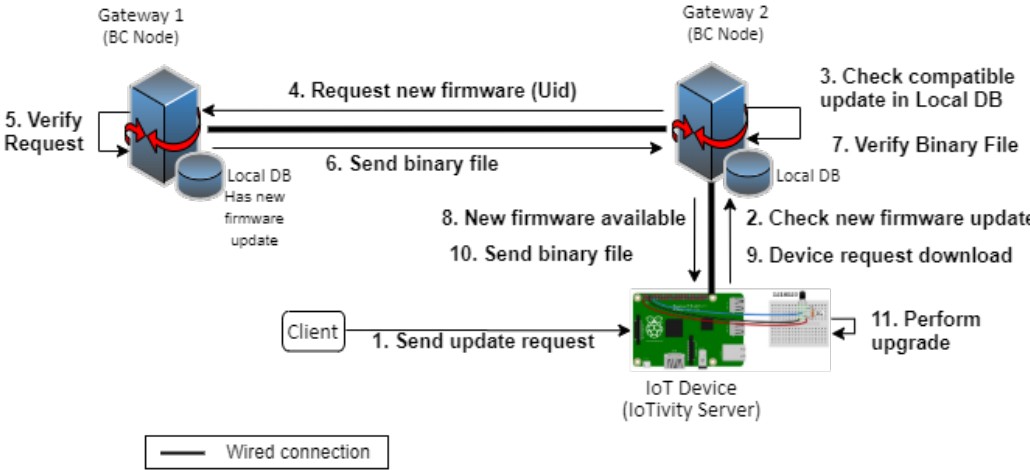

**Figure 9.** Implementation of a peer-to-peer firmware update distribution.

## 6. Evaluation and Discussion

### 6.1. Evaluation

This part describes the evaluation results of the implementation of our proposed method. We tested our direct firmware update and peer-to-peer firmware update implementation. We measured the time taken when finishing some numbers of requests that were set up for testing. We also measured the average latency and the average throughput of both the Check New Version step and Download Update step. For this test, we utilized the Autocannon, which is an HTTP benchmarking tool [30]. It provides two ways of usage, with the command line or programmatically using Node.js programming language. It also offers several parameters that we could configure according to our test purposes.

We evaluated our test in ten groups of requests; each group's total requests were increased by one hundred. It started with 100 requests, and the last group had 1000 requests. Table 4 shows the time performance for the Check New Version step. For the lowest number of requests, the direct firmware update protocol needs 4.28 s. To finish the last group, 1000 requests, this protocol needs 45.28 s. For peer-to-peer firmware update, the lowest number of requests, 100 requests, takes 5.22 s. The last group, 1000 requests, takes 36.26 s. Figure 10 shows the time performance for the Check New Version step of direct and peer-to-peer firmware update. The peer-to-peer firmware update protocol is faster than the direct firmware update protocol.

**Table 4.** Direct firmware update and peer-to-peer firmware update: the results of duration for the Check New Version step.

| Number of Requests | Duration (s) | |
| --- | --- | --- |
| | Direct Firmware Update | Peer-to-peer Firmware Update |
| 100 | 4.28 | 5.22 |
| 200 | 9.26 | 8.30 |
| 300 | 12.27 | 11.23 |
| 400 | 20.22 | 16.32 |
| 500 | 26.30 | 20.23 |
| 600 | 28.24 | 21.33 |
| 700 | 35.26 | 25.28 |
| 800 | 36.24 | 29.26 |
| 900 | 38.26 | 31.28 |
| 1000 | 45.28 | 36.26 |

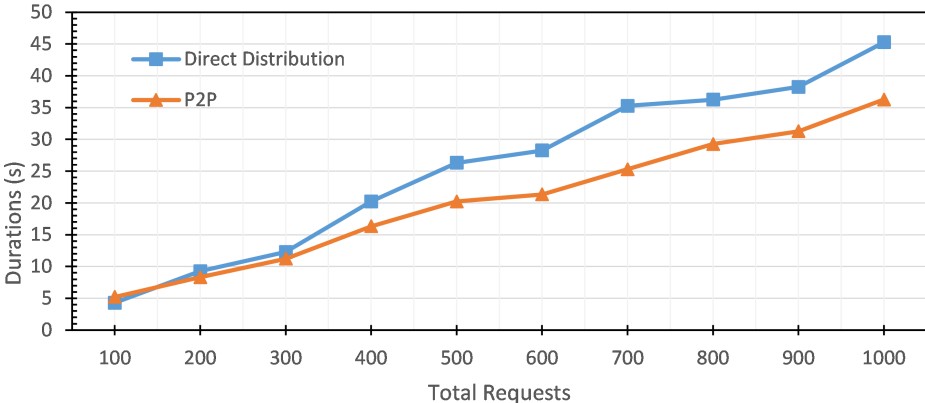

**Figure 10.** The duration of the Check New Version step: direct firmware update vs. peer-to-peer firmware update.

Figure 11 shows the comparison results of the average throughput and average latency performances of ten groups of requests in the Check New Version step for both protocols. Figure 11a shows that peer-to-peer firmware update protocol can handle 26.32 req/s. This value is higher than that of direct firmware update protocol. If there are 10,000 requests, the estimation time of peer-to-peer firmware update protocol to finish this step is 379.83 s or 6.3 min. The direct firmware update protocol can handle 22.1 req/s. That means that if there are 10,000 requests, the estimation time to finish this step is 452.38 s or 7.5 min. Figure 11b shows that the average latency of peer-to-peer firmware update is 360.92 ms. This value is lower than that for the direct firmware update protocol—437.52 ms. In terms of average latency and average throughput performance, the peer-to-peer protocol performs better than the direct update. The former communicates locally between gateways, while the latter communicates remotely between gateway and manufacturer.

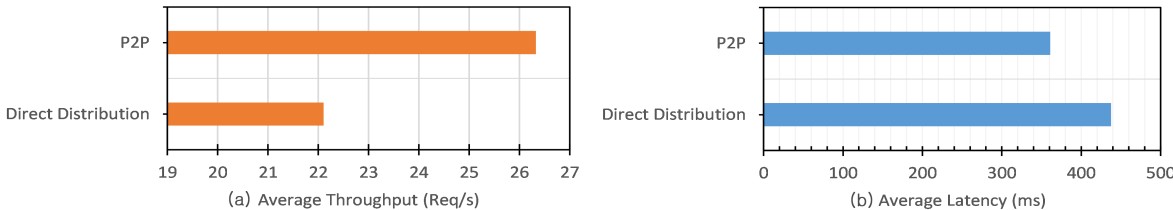

**Figure 11.** Comparison of average throughput (**a**) and average latency (**b**) in the Check New Version step between direct firmware update and peer-to-peer firmware update.

As the second part of the performance measurement, we measured the duration performance, the average latency, and the average throughput for the Download Update step. We evaluated with the same procedures as those for the Check New Version step. Table 5 shows the testing results of duration performance for the Download Update step of both direct firmware update and the peer-to-peer firmware update protocol. For the lowest request, the direct firmware update protocol needs 3.28 s, and for the last group, 1000 requests take 21.32 s. For peer-to-peer firmware updates, it takes 3.25 s to finish 100 requests, and it takes 19.23 s to finish 1000 requests. Figure 12 shows that the durations of both protocols to finish the second step are almost the same. However, the peer-to-peer protocol has a slightly lower duration than the direct firmware update protocol.

**Table 5.** Direct firmware update and peer-to-peer firmware update: the results of duration for the Download Update step.

| Number of Requests | Duration (s) | |
| --- | --- | --- |
| | Direct Firmware Update | Peer-to-Peer Firmware Update |
| 100 | 3.28 | 3.25 |
| 200 | 5.30 | 4.25 |
| 300 | 7.35 | 6.25 |
| 400 | 9.20 | 8.28 |
| 500 | 10.31 | 11.26 |
| 600 | 12.28 | 12.24 |
| 700 | 14.30 | 13.33 |
| 800 | 16.28 | 15.34 |
| 900 | 18.29 | 17.27 |
| 1000 | 21.32 | 19.23 |

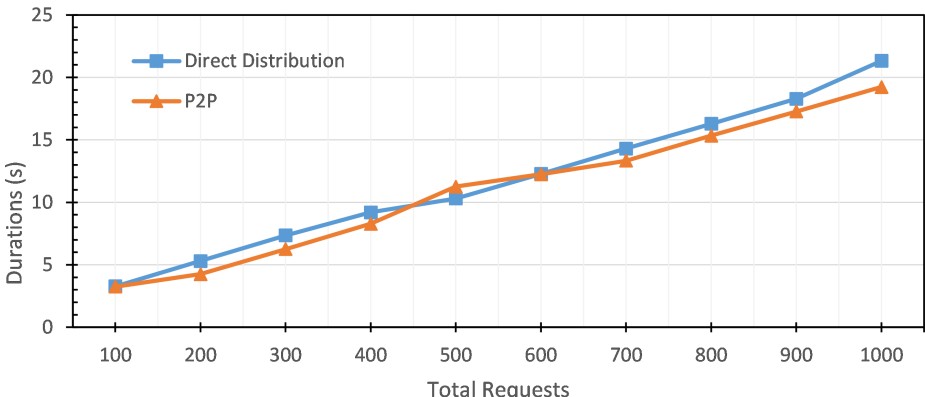

**Figure 12.** The duration of Download Update step: direct firmware update vs. peer-to-peer firmware update.

The average throughput and average latency of ten groups of requests are shown in Figure 13. The peer-to-peer protocol can handle 49.1 req/s. That means that if there are 10,000 requests, the peer-to-peer firmware update protocol can finish this step in 203.42 s or 3.3 min. This value is smaller compared to the direct firmware update protocol that can handle 45.8 req/s. That means that if there are 10,000 requests, the direct firmware update protocol can finish this step in 218.21 s or 3.6 min. The direct firmware update takes 11 min to finish 10,000 requests of two primary steps, and peer-to-peer takes 9.6 min. The average latency result for direct firmware update was 202.73 ms. This value is higher compared to the peer-to-peer protocol, which got a value of 189.58 ms. Our evaluations of both protocols in the Check New Version step and Download Update step show that the peer-to-peer firmware update protocol has better time performance than the direct firmware update protocol.

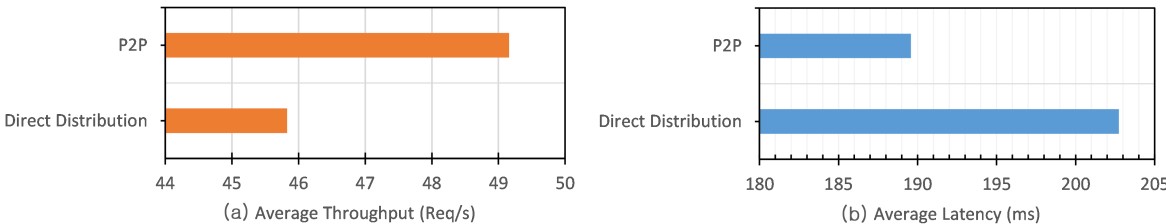

**Figure 13.** Comparison of average throughput (**a**) and average latency (**b**) in the Download Update step between direct firmware update and peer-to-peer firmware update.

We also evaluated the computational complexity and data transmission size for direct firmware update and peer-to-peer firmware update. As shown in Table 6, compared to the direct firmware

update, a peer-to-peer firmware update requires the extra computation of one hashing, one digital signature, and one signature verification. Additionally, it requires 197 bytes of extra data transmission. One gateway will need an extra 7.85 KB of memory space to implement a peer-to-peer update. With this cost, the peer-to-peer firmware update can increase overall firmware update availability and disseminate the traffic burden from the manufacturer's server to the gateways.

**Table 6.** Comparison of computational complexity and data transmission size between direct firmware update and peer-to-peer firmware update.

| Parameters | Direct Firmware Update | Peer-to-Peer Firmware Update |
| --- | --- | --- |
| Code size in gateway (KB) | 8.86 | 7.85 |
| Total # of Hash | 2 | 3 |
| Total # of Digital Signature | 2 | 3 |
| Total # of Signature Verification | 2 | 3 |
| Total data transmission size (bytes) | 455 | 652 |

*6.2. Discussion*

We can consider two cases of a single-point-of-failure problem for a firmware update distribution server. First, when the adversary disables the manufacturer's server service. Second, when the adversary successfully modifies the firmware update. For the first case, our proposed protocol has two ways to distribute the firmware update. Thus, if at least one of the gateways has downloaded the firmware before the manufacturer has disabled it, a device can get the firmware update by downloading the firmware from the gateway with the peer-to-peer firmware update. However, without this gateway's participation, we cannot provide peer-to-peer updates. To encourage gateways' participation, manufacturers can give gateways an incentive when they successfully participate in peer-to-peer update distribution. We consider this incentive mechanism viable future work. For the second case, if an attacker wants to modify the firmware update without being detected, the attackers should have to successfully modify the firmware update and the blockchain network's transaction block. To do that, the attacker needs to control more than 50% of the blockchain network. However, it is very difficult and costly to have that amount of computing power. Thus, our proposed protocol can overcome this kind of attack.

References [21,22] proposed the idea of blockchain-based firmware update distribution. In [21], their method gives some possibility to check the firmware version before the update begins, regarding which one has a higher or lower version. However, this will later cause redundancy checking that may cause unnecessary network traffic and increase nodes' computational power consumption and operations. Besides, the method only provides showcases for firmware updates from one manufacturer instead of the various manufacturers. In fact, many IoT devices from various manufacturers will exist in an IoT environment. In contrast, our protocols support multiband firmware updates because our protocol is based on the OCF specification that supports the various IoT devices category [31]. Reference [22] has innocuousness checking nodes that check the update against bugs and known attacks every time the manufacturer releases an update. Nevertheless, before the manufacturer releases a new firmware update in our proposed scheme, a trusted party (such as certified cybersecurity companies) must have already checked it. This process will shorten the firmware updates' distribution time.

These previous two solutions use Bitcoin technology. It takes around 10 min to verify a transaction and create a new block in Bitcoin—however, our proposed protocol is based on Ethereum. For Ethereum, the average time for one block creation is 10–20 s [23]. Time is an essential factor when distributing a newer version of firmware to the IoT devices. Moreover, developers can utilize smart contracts in Ethereum to build a decentralized application. The manufacturer will store each firmware update's metadata in the smart contract. When devices want to download the new firmware, they can verify it through the smart contracts. Furthermore, Ethereum provides a lightweight node

for constrained and low capacity devices that cannot afford to store multiple dozen Gigabytes of blockchain data [24,25].

In [26], the author also proposed a firmware update protocol based on blockchain, and it was designed based on Ethereum. The components in this protocol send the messages without any integrity protection. However, our proposed protocol provides the messages with integrity protection with a digital signature and hash algorithm. That is crucial because it will avoid the adversary from altering the messages and injecting malicious code during transmission.

## 7. Conclusions

We proposed two protocols to distribute a firmware update: direct firmware update and peer-to-peer firmware update distribution. Both protocols check the firmware update to a smart contract. However, the direct firmware update protocol downloads the update directly from the manufacturer's server. The peer-to-peer firmware update protocol happens between devices with the same specifications and from the same manufacturer. Therefore, IoT devices can share the available newest updates through IoT gateways. As the gateways provide update distribution on behalf of the manufacturer, single-point-of-failure is prevented, and firmware update availability is increased. For the blockchain network, we use Ethereum because the creation speed is faster than Bitcoin, which is around 10–20 s on average to create one block. Besides, Ethereum can utilize smart contracts to build a decentralized application because smart contracts will execute when the conditions are met. In the case of firmware update's distribution, each version of firmware released by the device's manufacturer will be recorded in the smart contract. When devices want to download the new firmware, they can verify it through the smart contract. Furthermore, Ethereum provides a lightweight node for constrained and low capacity devices that cannot afford to store multiple dozen Gigabytes of blockchain data.

We implemented our proposed protocols in Raspberry Pi 3 Model B and the VMWare virtual environment. There are three nodes in the blockchain network: one manufacturer and two gateways. We evaluated the proposed system's performance by measuring the average latency, average throughput, and duration of both protocols to finish two main steps, Check New Version step and Download Update step. With request/second values, we measured the total duration that it takes to handle 10,000 requests. The direct firmware update needs 11 min, and the peer-to-peer firmware update needs 9.6 min. These evaluations show that the peer-to-peer firmware update protocol has better time performance than the direct firmware update protocol. We also evaluated the computational complexity and data transmission size for the direct firmware update and the peer-to-peer firmware update. Compared to the direct firmware update, the peer-to-peer firmware update requires the extra computation of one hashing, one digital signature, and one signature verification. Additionally, it requires 197 bytes of extra data transmission. One gateway will need an extra 7.85 KB of memory space to implement a peer-to-peer update. With this cost, the peer-to-peer firmware update can increase overall firmware update availability and disseminate the manufacturer's server's traffic burden to the gateways.

**Author Contributions:** Conceptualization, E.N.W. and Y.E.O.; data curation, E.N.W. and Y.E.O.; formal analysis, E.N.W. and Y.E.O.; funding acquisition, S.-G.L. and J.-H.L.; investigation, E.N.W. and Y.E.O.; methodology, E.N.W. and Y.E.O.; project administration, S.-G.L. and J.-H.L.; resources, S.-G.L. and J.-H.L.; software, E.N.W. and Y.E.O.; supervision, S.-G.L. and J.-H.L.; validation, E.N.W. and Y.E.O.; visualization, E.N.W.; Writing—original draft, E.N.W.; writing—review and editing, E.N.W., Y.E.O., S.-G.L., and J.-H.L. All authors have read and agreed to the published version of the manuscript.

**Funding:** This work was supported by Basic Science Research Program through the National Research Foundation of Korea (NRF) funded by the Ministry of Education (grant number: 2018R1D1A1B07047601) and also supported by Dongseo University Research Fund of 2020.

**Acknowledgments:** We would like to thank the anonymous reviewers for their comments and suggestions that helped us improve the paper.

**Conflicts of Interest:** The authors declare no conflict of interest.

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
