# Peer review of "A Blockchain-Based OCF Firmware Update for IoT Devices"

_applsci, doi:10.3390/app10196744_

Round 1

Reviewer 1 Report

My major question concerns the peer-to-peer update mechanism, and the question whether an average gateway will have enough processing power, memory and bandwith to act as a full blockchain node.

Please, check language for spelling and especially grammar.

More specific feedback:

- Line 23 - The Cisco prediction dates from 2013(?), it does not make a lot of sense to cite it in 2020.
- Line 42f. - A widely used update process involves manual updates by the user. Connecting a device to a PC and downloading updates thru the PC.
- Line 192ff - From my point of view and with the understanding that I got while reading until that point, I do not see a difference in Bitcoin and Ethereum. Both - at the time of writing - rely on proof of work algorithms, and running a full node (for verification of transactions) on an IoT device might in most cases not be feasible in terms of processing power, bandwith and memory for both.
- Table 1: Is purl the same as url (in the text above the table)?
- Figure 2: As far as I understand, the gateway nodes are smart home gateways, for example. At the same time they need to be full nodes on the blockchain. As discussed above the hardware requirements for full nodes are quite high, even in Ethereum. Is it realistic to have smart home gateways acting as full nodes on a blockchain?
- Chapter 5.2.2 - Line 399 - Does the device initiate the check for updates? Is there a way to push updates to the devices, without the devices checking?

Author Response

Response to Reviewer 1 Comments

Please see the attachment. We give the compared version of our manuscript.

My major question concerns the peer-to-peer update mechanism, and the question whether an average gateway will have enough processing power, memory and bandwidth to act as a full blockchain node.

→ Ethereum provides a lightweight node for low-capacity devices. A light client can be viewed as a client that downloads block headers by default, and verifies only a small portion of what needs to be verified, using a distributed hash table as a database for trie nodes in place of its local hard drive. However, this node will have limited stored blocks history and also the speed is slower compared to full nodes. Light nodes process about 1KB of data per 2 minutes to receive data from the network.

References: https://docs.ethhub.io/using-ethereum/running-an-ethereum-node/

        https://eth.wiki/en/concepts/light-client-protocol

Please, check language for spelling and especially grammar.

→ Yes, we’ve checked it using the grammar checking software.

More specific feedback:

- Line 23 - The Cisco prediction dates from 2013(?), it does not make a lot of sense to cite it in 2020.

→ changed the Cisco reference to another reference.

- Line 42f. - A widely used update process involves manual updates by the user. Connecting a device to a PC and downloading updates thru the PC.

→ added some sentences to line 42-43.

- Line 192ff - From my point of view and with the understanding that I got while reading until that point, I do not see a difference in Bitcoin and Ethereum. Both - at the time of writing - rely on proof of work algorithms, and running a full node (for verification of transactions) on an IoT device might in most cases not be feasible in terms of processing power, bandwidth and memory for both.

→ I’ve explained more about the difference between bitcoin and ethereum. I also added explanation about how the gateway can be feasible in the blockchain network. Ethereum provides the lightweight node to support the device with limited memory and processing power. Line 198-206.

- Table 1: Is purl the same as url (in the text above the table)?

→ Yes, PURL and URL are the same. PURL stands for Package URL that is used by OCF. It refers to a link to download the new firmware.

- Figure 2: As far as I understand, the gateway nodes are smart home gateways, for example. At the same time they need to be full nodes on the blockchain. As discussed above the hardware requirements for full nodes are quite high, even in Ethereum. Is it realistic to have smart home gateways acting as full nodes on a blockchain?

→ Refer to our answer in the first comment.

- Chapter 5.2.2 - Line 399 - Does the device initiate the check for updates? Is there a way to push updates to the devices, without the devices checking?

Yes, it does. Our implementation is based on the OCF firmware update protocol. So far, OCF specification does not provide push updates to the devices. OCF only provides device initiated updates.

Reviewer 2 Report

1: Which "devices growth rate" are authors talking about? Is it device adoption, device installation, device development, device deployment or some other growth rate? Please, focus this appropriately.

2-3: Firmware update is not related to development, as it relies on the existing hardware base. Please, rephrase and if appropriate, focus onto continuous development of the software part of the device.

3: "to the distributor, retailer, and customer" is unnecessary repetition, and can be omitted.

5: Overall comment: please, proofread the text once again by a skilled English speaker. For example, standalone sentence "It is a centralized way and direct downloads

from the manufacturer server." has little meaning.

7: "Needs to ensure" instead of "needs to make sure".

8-12: Please omit general description of the blockchain from abstract as it is widely known and recognized. For example, it is not necessary to describe in the abstract that blockchain is rapidly developing, can be potential solution to security problems, that its transactions can be verified by all participants, that information cannot be erased etc.

15: "we propose" instead of "we proposed".

17-18, and elsewhere: The authors claim that their proposal solves the issue of single point of failure. However, what if the blockchain based firmware distribution is attacked? Wouldn't it hinder update process and is that a single point of failure? Please recognize this as a downside to the proposed system somewhere in the paper.

22: Please change "The concept of IoT is connecting the unconnected" to a more appropriate "networking of previously not connected devices", or something similar

22-23: "By now, around 14 billion of "things" are 23 connected to the internet. Cisco Systems predicts that by 2020, this number will reach 50 billion." The paper is submitted for review mid-Aug 2020. This reference needs to include more recent prediction, that refers to the period in the future, and not 2020. that is already nearing its end.

28-29: the same remark as for 3:.

29: Please insert "Needs" instead of "need" in "The aspects that need".

32-39: If these notions are cited from somewhere, they need to be referenced. If not, skip this comment.

40: Rephrase "The other important thing".

40: Rephrase "we need" to an impersonal statement.

48-50: Sentence "OCF not only provide interoperability services between IoT devices, but they also provide documentation for getting the firmware updates to the OCF device from the external server, which is the manufacturer." is meaningless. Check and rephrase accordingly.

51-52: "With this limited protocol interactions, there is no step to ensure the integrity of the package." This claim is simply not correct. Why the driver package could not be simply signed and then checked for consistency? Windows OS has been using the same process for more than a decade.

54-65: What about extremely limited processing, memory and other resources in SoC IoT systems? The authors never recognize this as being an obstacle for adoption of their proposal. Under current development of technology, and taking this into consideration, would implementation of blockchain be feasible at all? Please, comment this fact extensively in the paper.

82-83: If this claim is cited, it should be referenced.

84-85: "Blockchain creates permanent 85 records and histories of transactions, but nothing is really permanent." should be rephrased, as it is contradictory in its own terms.

88: Sentence "Whoever in the blockchain network can observe" needs to be rephrased.

89: "This verification is useful for the prevention of currency "double-spending" - please explain in a few words what currency double spending is. Why is it mentioned anyway in context of this paper that is not dealing with digital currencies?

92-93: The following sentence cannot stand alone: "Two properties required for

digital signature scheme [7]."

99-100: The following sentence "This function should have properties of a deterministic, one-way function, collision resistance, and pseudorandom." needs significant rephrasing as it is currently meaningless.

103: Do not start sentence with "and".

107 on: Why do authors suddenly start explaining Ethereum and smart contracts? It is not mentioned until now, neither in Abstract nor in Introduction. Do the authors want to base their firmware update system on Ethereum? If so, this needs to be properly announced beforehad and throughout the paper until this poiint.

109-111 on: The authors are not writing about digital currencites. Why mentioning cryptocurrencies and Bitcoin at all in this paper? All references to cryptocurrencies should be omitted from the paper as they could be misleading for the reader.

107-130: This entire paragraph should be deleted because it is not connected with the rest of the paper. It explains Ethereum technology.

134: "It is called middleware." What is called middleware?

143: Delete entire sentence "We will explain briefly about the OCF firmware update protocol from Figure 1.".

145: "In order to explain our proposed protocol in Section 4". The protocol needs to be explained earlier in the paper in order to be referenced, and not later. Please remedy this.

152: "Maintain" instead of "maintains".

132-167: The paragraph is titled "OCF", but it explains upgrade process.

170: Figure 1 is splitting underlying sentence in half without being properly announced.

176-177: "With this limited protocol interactions, it means that there is no step to ensure the integrity of the package." Why are authors adding a significant layer of complexity by introducing blockchain technology instead of simply trying to modify the protocol to include checking of firmware package signature?

192: "These previous two solutions use bitcoin-based technology" This is a strange statement. Blockchain or bitcoin?

197: Authors are again discussion Ethereum. Then, in line 203, they finally state that they are proposing blockchain based firmware update. This further underlines, just like already said for line 107:, that all references made to Ethereum and especially explanations, should be deleted.

203: "In our proposed scheme, we propose two update protocols" Please change repetitive verbal forms.

206: "By providing these two ways to update the firmware, the burden of network traffic will 207 be distributed from server to blockchain through the peer-to-peer update scheme so it will increase 208 overall update availability." What about downsides of the approach? Increased bandwidth and utilization of other resources (CPU, memory). Is there impact on power saving scheme of IoT devices? The authors are advised to dedicate a certain portion of the paper to downsides of their proposal, but also possibilities of further research.

213: Please check and rephrase the following sentence: "Protocols in [24] are a direct and indirect firmware update."

216-221: All statements in the following section need to be checked and rephrased, both from semantic and syntax points of view, including internal logic: "Besides, our protocol provides two ways to download the update as already mentioned before. Moreover, in, their component sending themessages without any cryptographic algorithm protection, and also they receive the messages without verifying the sender. Contrast to our proposed protocol that provides the messages with a digital signature and hash algorithm for verification. It is crucial because it will avoid the adversary to alter the messages and inject the malicious code."

223: "Some components that are contribute" Please check language. Again, use informal forms, there is no "we".

224: "Outside of the blockchain network, there are the manufacturers" Subsequently, in 232, "Manufacturer Node. A node inside of the blockchain network owned by the manufacturer." These two statements are fully in contradiction.

240: "IoT gateway can be a Wi-Fi router in a smart home, and it will be inside of the

blockchain." The authors are implying that their system of firmware update is aimed towards simplest IoT applications like home automation. Furthermore, prior to this, they mention smart bulbs etc. If so, this should be stated very early in the paper, and in the abstract. So far, until line 240, it seems as if the authors are dealing with industrial and complex applications, while from this line, it seems that the authors for the first time claim home applications. This changes course of the paper fully, and it should be reflected in the title, introduction and other key paragraphs of the paper.

250: "It stores" instead of "It stores".

254: Table 1 should be place after line 248, or its announcement should be moved just prior to the table.

257-261: What is the downside to this approach? What about impact on the cost of IoT devices?

263-264, 266: Grammatically completely incorrect, please remedy: "There are four entities that involve in this protocol. Which are OCF device, gateway node or IoT gateway, firmware update smart contract, and the manufacturer." - "We put the detail explanation below:"

284: "If they are equal" instead of "If the same,"

292: "After the initialization finished" should be changed to "After the initialization is finished"

293: "After package complete" is incorrect.

298: install OF the new firmwares

301: The statement "Peer-to-peer firmware update protocol happens between devices of the same type and from the same manufacturer." is not correct. Devices of the same type and manufacturer, but different generation regularly have different firmwares.

304 - end of the paper; numerous grammar and semantic errors, too many to enumerate all within the review

343: "5. Implementation and Analysis" - analysis of what?

355: "Raspbian Buster", not "Raspbian buster".

365, 372, 420 and elsewhere in the paper: "Entities" instead of "entity".

365: What does sentence "The manufacturer and blockchain network that programmed in Virtual Machine." mean?

414: Sentence "This step happened inside the IoT device" should be rephrased using more suitable terms.

418: Figures 7. and 8. should not be inserted in sequence without text between them.

434: Change title from "Evaluation and Analysis" to "Discussion"

477- Figures should not be inserted in sequence without text between them. Major drawback of the paper is also the fact that the authors discuss security in the beginning but at the end, they only measure update speed without any reference to security in the paper? There is absolutely no analysis of quality of the described and proposed approach in comparison to existing technology.

486: The conclusion is too short, contains very few details and needs to be significantly expanded, both in quantity and quality, after acknowledging the rest of reviewer's comments.

Author Response

Response to Reviewer 2 Comments

Please see the attachment. We give the compared version of our manuscript.

1: Which "devices growth rate" are authors talking about? Is it device adoption, device installation, device development, device deployment or some other growth rate? Please, focus this appropriately.

→ Rephrased the sentence in line 1.

 2-3: Firmware update is not related to development, as it relies on the existing hardware base. Please, rephrase and if appropriate, focus onto continuous development of the software part of the device.

→ Rephrased the “IoT device development” to “IoT device software development”.

 3: "to the distributor, retailer, and customer" is unnecessary repetition, and can be omitted.

→ Removed “distributor, retailer”.

 5: Overall comment: please, proofread the text once again by a skilled English speaker. For example, standalone sentence "It is a centralized way and direct downloads from the manufacturer server." has little meaning.

→ Rephrased the sentence line 5 to “OTA happened in a centralized way and directly downloads the firmware update from the manufacturer’s server.”

 7: "Needs to ensure" instead of "needs to make sure".

→ Changed the words.

 8-12: Please omit general description of the blockchain from abstract as it is widely known and recognized. For example, it is not necessary to describe in the abstract that blockchain is rapidly developing, can be potential solution to security problems, that its transactions can be verified by all participants, that information cannot be erased etc.

→ Removed the blockchain explanation.

 15: "we propose" instead of "we proposed".

→ Changed to “we propose”.

 17-18, and elsewhere: The authors claim that their proposal solves the issue of single point of failure. However, what if the blockchain based firmware distribution is attacked? Wouldn't it hinder update process and is that a single point of failure? Please recognize this as a downside to the proposed system somewhere in the paper.

→ We can consider two cases of a single point of failure problem for a firmware update distribution server. First, when the adversary disabled the manufacturer’s server service. Second, when the adversary successfully modifies the firmware update. For the first case, our proposed protocol has two ways to distribute the firmware update. So, if one of the gateways has downloaded the firmware before the manufacturer has disabled, the devices can get the firmware update by downloading the firmware from the gateway with peer-to-peer firmware update. If none of the gateways has downloaded the firmware, we cannot solve this problem. For the second case, if an attacker wants to modify the firmware update without being detected, the attackers should have to successfully modify the firmware update and the transaction block in the blockchain network. In order to do that, the attacker needs to control more than 50% of the blockchain network. However, it is very hard and costly to have so much computing power. So, our proposed protocol can overcome this kind of attack.

 22: Please change "The concept of IoT is connecting the unconnected" to a more appropriate "networking of previously not connected devices", or something similar

→ Rephrased the sentence line 22

 22-23: "By now, around 14 billion of "things" are 23 connected to the internet. Cisco Systems predicts that by 2020, this number will reach 50 billion." The paper is submitted for review mid-Aug 2020. This reference needs to include more recent prediction, that refers to the period in the future, and not 2020. that is already nearing its end.

Changed the Cisco reference to another reference.

 28-29: the same remark as for 3:.

→ Removed “distributor, retailer”.

 29: Please insert "Needs" instead of "need" in "The aspects that need".

→ Changed the word.

 32-39: If these notions are cited from somewhere, they need to be referenced. If not, skip this comment.

→ These notions are not cited from the other reference.

 40: Rephrase "The other important thing".

→ Rephrased the sentence line 39. “Based on several reasons mentioned above”

 40: Rephrase "we need" to an impersonal statement.

→ Rephrased the sentence line 40.

 48-50: Sentence "OCF not only provide interoperability services between IoT devices, but they also provide documentation for getting the firmware updates to the OCF device from the external server, which is the manufacturer." is meaningless. Check and rephrase accordingly.

→ Rephrased the sentence line 48-50. “OCF as one of the organizations that supports IoT devices interoperability also provides a firmware update protocol.”

 51-52: "With this limited protocol interactions, there is no step to ensure the integrity of the package." This claim is simply not correct. Why the driver package could not be simply signed and then checked for consistency? Windows OS has been using the same process for more than a decade.

The context of that sentence is talking about the limited step in the OCF firmware update protocol between IoT devices and manufacturers (Refer to figure 1). When the firmware update distribution process happens, the IoT devices want to check the new firmware update and download the firmware update, but  there is no step to verify the integrity of the information and the package in the OCF protocol.

 54-65: What about extremely limited processing, memory and other resources in SoC IoT systems? The authors never recognize this as being an obstacle for adoption of their proposal. Under current development of technology, and taking this into consideration, would implementation of blockchain be feasible at all? Please, comment this fact extensively in the paper.

→ Yes, it is feasible. Ethereum provides a lightweight node for low-capacity devices. A light client can be viewed as a client that downloads block headers by default, and verifies only a small portion of what needs to be verified, using a distributed hash table as a database for trie nodes in place of its local hard drive. However, this node will have limited stored blocks history and also the speed is slower compared to full nodes. Light nodes process about 1KB of data per 2 minutes to receive data from the network.

References: https://docs.ethhub.io/using-ethereum/running-an-ethereum-node/

        https://eth.wiki/en/concepts/light-client-protocol

 82-83: If this claim is cited, it should be referenced.

→ Cited this paragraph.

 84-85: "Blockchain creates permanent 85 records and histories of transactions, but nothing is really permanent." should be rephrased, as it is contradictory in its own terms.

→ Rephrased the sentence line 86-87. “Blockchain creates permanent records and histories of transactions.”

 88: Sentence "Whoever in the blockchain network can observe" needs to be rephrased.

→ Rephrased the sentence line 84-85. “Whoever in the blockchain network can monitor and verify the authenticity of the transactions.”

 89: "This verification is useful for the prevention of currency "double-spending" - please explain in a few words what currency double spending is. Why is it mentioned anyway in context of this paper that is not dealing with digital currencies?

Removed this sentence to prevent misleading for the reader.

 92-93: The following sentence cannot stand alone: "Two properties required for digital signature scheme [7]."

→ Rephrased the sentence line 95-96. “There are two properties required for the digital signature scheme.“

 99-100: The following sentence "This function should have properties of a deterministic, one-way function, collision resistance, and pseudorandom." needs significant rephrasing as it is currently meaningless.

→ Rephrased the sentence line 107-114.

 103: Do not start sentence with "and".

→ Removed “and”.

 107 on: Why do authors suddenly start explaining Ethereum and smart contracts? It is not mentioned until now, neither in Abstract nor in Introduction. Do the authors want to base their firmware update system on Ethereum? If so, this needs to be properly announced beforehad and throughout the paper until this poiint.

Actually already mentioned about smart contract in the abstract. Yes, for the implementation of our protocol, we use Ethereum and smart contract. Now I modified and mentioned in the abstract about Ethereum-based technology that we used in our implementation.

109-111 on: The authors are not writing about digital currencites. Why mentioning cryptocurrencies and Bitcoin at all in this paper? All references to cryptocurrencies should be omitted from the paper as they could be misleading for the reader.

Removed description related to digital currency to prevent misleading for the reader.

 107-130: This entire paragraph should be deleted because it is not connected with the rest of the paper. It explains Ethereum technology.

Removed some sentences related to ethereum and cryptocurrency but kept some sentences  that related with smart contract because we used it in our implementation.

 134: "It is called middleware." What is called middleware?

→ Removed this sentence.

 143: Delete entire sentence "We will explain briefly about the OCF firmware update protocol from Figure 1.".

→ Removed the sentence.

 145: "In order to explain our proposed protocol in Section 4". The protocol needs to be explained earlier in the paper in order to be referenced, and not later. Please remedy this.

→ Rephrased the sentence line 151-153. “For the convenient explanation of our proposed protocol, we add two notes that are called ‘new firmware available’ and ‘software validation’ into the original OCF firmware update protocol. ”

 152: "Maintain" instead of "maintains".

→ Changed the word.

 132-167: The paragraph is titled "OCF", but it explains upgrade process.

Changed the title to OCF Firmware Update Protocol.

 170: Figure 1 is splitting underlying sentence in half without being properly announced.

Added “from top-left to bottom-right” in the caption.

 176-177: "With this limited protocol interactions, it means that there is no step to ensure the integrity of the package." Why are authors adding a significant layer of complexity by introducing blockchain technology instead of simply trying to modify the protocol to include checking of firmware package signature?

To ensure the integrity of the package we can simply modify the protocol by including checking of the firmware package signatures. But, we introduced blockchain in the OCF firmware update protocol. By using blockchain technology, peer-to-peer firmware updates become possible. So, between IoT devices can share the available newest update. Then, they can check the integrity and validity of the update to the blockchain network. Besides, the burden of the traffic will be moved from the manufacturer to the blockchain network and it will prevent single-point-of-failure and provide firmware update availability.

 192: "These previous two solutions use bitcoin-based technology" This is a strange statement. Blockchain or bitcoin?

→ Yes, they use bitcoin, and bitcoin is one of the blockchain implementations.

 197: Authors are again discussion Ethereum. Then, in line 203, they finally state that they are proposing blockchain based firmware update. This further underlines, just like already said for line 107:, that all references made to Ethereum and especially explanations, should be deleted.

Ethereum is one of the implementations of blockchain technology. In this paper we used Ethereum. That's why we discuss Ethereum.

 203: "In our proposed scheme, we propose two update protocols" Please change repetitive verbal forms.

→ Rephrased the sentence line 212. “In this section we proposed blockchain-based OCF firmware update protocols. The update protocols consist of two ways of distributing firmware update: direct and peer-to-peer update.”

 206: "By providing these two ways to update the firmware, the burden of network traffic will 207 be distributed from server to blockchain through the peer-to-peer update scheme so it will increase 208 overall update availability." What about downsides of the approach? Increased bandwidth and utilization of other resources (CPU, memory). Is there impact on power saving scheme of IoT devices? The authors are advised to dedicate a certain portion of the paper to downsides of their proposal, but also possibilities of further research.

Comparison of Computational Complexity and Data Transmission Size between Direct Firmware Update and Peer-to-peer Firmware Update

Direct Firmware Update

Peer-to-peer Firmware Update

Code size in gateway (KB)

8.86

7.85

Total # of Hash

2

3

Total # of Signaturing

2

3

Total # of signature verification

2

3

Total data transmission size (bytes)

455 

652

Compared to direct firmware update, peer-to-peer firmware update requires extra computation of 1 hashing, 1 signaturing, and 1 signature verification. Also, it requires 197 bytes extra data transmission. One gateway will need an extra 7.85 kB of memory space to implement peer-to-peer update. With this cost, the peer-to-peer firmware update can increase the availability of firmware updates and distribute the traffic burden of the manufacturer’s server to the gateways.

One of the main drawbacks of peer-to-peer update is it is only possible if at least one of the gateways downloaded the firmware and distributed the firmware to other nodes. However, without this gateway’s participation, we cannot provide peer-to-peer updates. To encourage gateways participation, manufacturers can give an incentive to gateways when they successfully participate in peer-to-peer updates distribution. We consider this incentive mechanism as one of our future work. We’ve updated our manuscript to address this issue in the Section Discussion.

 213: Please check and rephrase the following sentence: "Protocols in [24] are a direct and indirect firmware update."

→ Rephrased sentence line 222. “Protocols in [24] provide two types of blockchain-based firmware update protocols. One is initiated by the manufacturer’s server and another initiated by the IoT device. OCF protocol provides the IoT device initiated protocol. So, we implement only IoT device initiated protocol.”

 216-221: All statements in the following section need to be checked and rephrased, both from semantic and syntax points of view, including internal logic: "Besides, our protocol provides two ways to download the update as already mentioned before. Moreover, in, their component sending the messages without any cryptographic algorithm protection, and also they receive the messages without verifying the sender. Contrast to our proposed protocol that provides the messages with a digital signature and hash algorithm for verification. It is crucial because it will avoid the adversary to alter the messages and inject the malicious code."

Rephrased the sentences in this paragraph. “The components in protocol [24] send the messages without any integrity protection. But, our proposed protocol provides the messages integrity protection with a digital signature and hash algorithm. It is crucial because it will avoid the adversary to alter the messages and inject the malicious code during transmission.”

 223: "Some components that are contribute" Please check language. Again, use informal forms, there is no "we".

→ Rephrased the sentence line 230-231

 224: "Outside of the blockchain network, there are the manufacturers" Subsequently, in 232, "Manufacturer Node. A node inside of the blockchain network owned by the manufacturer." These two statements are fully in contradiction.

Rephrased sentence line 232-248.

 240: "IoT gateway can be a Wi-Fi router in a smart home, and it will be inside of the blockchain." The authors are implying that their system of firmware update is aimed towards simplest IoT applications like home automation. Furthermore, prior to this, they mention smart bulbs etc. If so, this should be stated very early in the paper, and in the abstract. So far, until line 240, it seems as if the authors are dealing with industrial and complex applications, while from this line, it seems that the authors for the first time claim home applications. This changes course of the paper fully, and it should be reflected in the title, introduction and other key paragraphs of the paper.

→ We only put that to give examples. Our firmware update protocol is based on the OCF. OCF supports many IoT devices category: smart home, appliances, electronics, etc. This is the link for OCF device specification https://openconnectivity.org/specs/OCF_Device_Specification_v2.2.0.pdf.

 250: "It stores" instead of "It stores".

→ Changed the word to “It stores”.

 254: Table 1 should be place after line 248, or its announcement should be moved just prior to the table.

The table already placed after 248 since the beginning

 257-261: What is the downside to this approach? What about impact on the cost of IoT devices?

→ The firmware update will be checked by the trusted party before this process begins. It will not affect our protocol’s flow. However, if the trusted party has not checked the released firmware and there is a problem in the update, it will delay the firmware update distribution.

 263-264, 266: Grammatically completely incorrect, please remedy: "There are four entities that involve in this protocol. Which are OCF device, gateway node or IoT gateway, firmware update smart contract, and the manufacturer." - "We put the detail explanation below:"

→ Rephrased the sentence line 267-270.

 284: "If they are equal" instead of "If the same,"

→ Corrected the sentence.

 292: "After the initialization finished" should be changed to "After the initialization is finished"

→ Corrected the sentence.

 293: "After package complete" is incorrect.

Changed to “After the OCF device successfully download the package, softwareupgrade resource will send SVA state to the pstat resource and client.”

 298: install OF the new firmwares

→ Changed to “After an OCF device successfully installed the new firmware, the pstat resource in the device will change the tm bit value to 0.”

 301: The statement "Peer-to-peer firmware update protocol happens between devices of the same type and from the same manufacturer." is not correct. Devices of the same type and manufacturer, but different generation regularly have different firmwares.

→ Rephrased the sentence. “Peer-to-peer firmware update protocol happens between devices with the same specification and from the same manufacturer.”

 304 - end of the paper; numerous grammar and semantic errors, too many to enumerate all within the review

→ Checked the grammar and semantic errors with related software.

 343: "5. Implementation and Analysis" - analysis of what?

→ Analysis of our evaluation. Changed the title to “Implementation” and separated the evaluation part.

 355: "Raspbian Buster", not "Raspbian buster".

→ Changed the word.

 365, 372, 420 and elsewhere in the paper: "Entities" instead of "entity".

→ Changed the word.

 365: What does sentence "The manufacturer and blockchain network that programmed in Virtual Machine." mean?

→ Rephrased sentence line 354-355.

 414: Sentence "This step happened inside the IoT device" should be rephrased using more suitable terms.

→ Rephrased the sentence line 411-421.

 418: Figures 7. and 8. should not be inserted in sequence without text between them.

→ Rearranged the text and figures.

 434: Change title from "Evaluation and Analysis" to "Discussion"

Separate “5. Implementation and Analysis” into two sections “5. Implementation” and “6. Evaluation and Discussion”. Under Section 6 has been separated into two subsections, “6.1 Evaluation” and “6.2 Discussion”.

 477- Figures should not be inserted in sequence without text between them. Major drawback of the paper is also the fact that the authors discuss security in the beginning but at the end, they only measure update speed without any reference to security in the paper? There is absolutely no analysis of quality of the described and proposed approach in comparison to existing technology.

→ The problem that our proposed protocols want to solve is the availability of the firmware update if single-point-of-failure happens. Time becomes one of the important parts during distributing the firmware update. So, in this paper we want to benchmark the speed of our protocol to see is it makes a difference or not between both protocols. Also, we put the security analysis in the new manuscript in Section Discussion.

 486: The conclusion is too short, contains very few details and needs to be significantly expanded, both in quantity and quality, after acknowledging the rest of reviewer's comments.

→ Added more details in the conclusion from both of reviewers' comments.

Round 2

Reviewer 2 Report

Please proofread the text, preferably by a skilled English speaker.

Author Response

Comments: Please proofread the text, preferably by a skilled English speaker.

Answer: We have proofread the text and edited the abstract, introduction, conclusion, and some minor revisions in the other sections. We highlighted the revised parts.